

# Debris flow impact estimation on a rigid barrier

Federico Vagnon[1] and Andrea Segalini[2]

[1]Department of Earth Sciences, University of Turin, Turin, 10135, Italy
[2]Department of Civil-Environmental Engineering and Architecture, University of Parma, Parma, 43124, Italy

*Correspondence to*: Federico Vagnon (fvagnon@unito.it)

**Abstract.** The aim of this paper is to analyse debris flow impact against rigid and undrained barrier in order to propose a new formulation for the estimation of acting force after the flow impact to safe design protection structures. For this reason this work concentrate on the flow impact, by performing a series of small scale tests in a specifically created flume. Flow

characteristics (flow height and velocity) and applied loads (dynamic and static) on barrier were measured using 4 ultrasonic devices, 4 load cells and a contact surface pressure gauge. The results obtained were compared with main existing models and a new equation is proposed. Furthermore, a brief review of the small scale theory was provided to analyse the scale effects that can affect the results.

**Keywords:** Granular materials; debris flows; impact force.

## 1 Introduction

Every year several gravitational movements, such as debris flows, landslides and avalanches, affect mountainous regions all over the world. Understanding and predicting their interactions with protection structures is a key point for the assessment and the management of risk.

Debris flow impact estimation requires to analyse and to discuss two key points: the first one is the data availability, deriving from real case observation, to validate the proposed models; the second one is related with the universal applicability of these equations.

Concerning the first point, the difficulties to find available data derived from monitoring of debris flow events force to perform laboratory experiments (Armanini and Scotton, 1992; Hubl and Holzinger, 2003; Canelli et al, 2012) due to the high

instrumentation costs.

Furthermore, laboratory tests allow to keep under strict control all the parameters involved and to easily perform several analyses. On the other hand, the question about the scale effects it is not completely solved: many authors raised doubts about the acceptability of the results carried out with these experiments (Iverson, 1997).

The design of mitigation structures requires simple models to predict impact pressure with high reliability; these models

should be universally recognized and should include few parameters, easy to estimate. Further, material properties and flow



characteristics should be considered in the equations. Following these preconditions, an accurate study of the flow behaviour against structures is necessary in order to define European criteria for design of debris flows protection fences.

This paper presents the first results of several laboratory tests performed to reduce the lack of information about impact force prediction.

## 2 Small scale theory

The possibility to simulate debris flows in laboratory is an argument debatable and controversial; even if the similarity theory provides the necessary support to design models and to extrapolate the data at the real scale, the scale effects plays an important role in the comprehension of the phenomenon (Longo, 2011). In particular, while geometric similarity ($\lambda$) can be easily obtained as the ratio of the prototype length ($L''$) and laboratory conditions ($L'$), the major limit is represented by dynamic similarity of all forces because are strictly related with the nature and the viscosity of the fluid (Iverson, 1997).

These issues were faced starting from an accurate dimensional analysis of the impact of a saturated mass against a rigid wall. The longitudinal deformation of this structure is the key parameter that allows to evaluate the energy dissipation of the mass. From this point of view, the mean density of saturated debris $\rho_m$, the flow velocity $v_f$, the flow height $h_f$ and the channel width $B$, play a fundamental role in the characterization of the mobilized mass. On the other hand, the impact and the debris retention behind the barrier are related to elastic module $E$, yield stress $\sigma_0$ and shape factor of the barrier. Thus, the maximum barrier deformation can be expressed as:

$$\delta = f(\rho_m, v, h_f, B, E, \sigma_0, \text{shape}), \tag{1}$$

The similarity criteria, produces these conditions:

$$\begin{cases} r_\delta = r_B \\ \lambda = r_B \\ r_{\sigma 0} = r_{\rho m} \cdot r_v^2 \\ r_E = r_{\rho m} \cdot r_v^2 \\ r_v = \sqrt{\lambda} \end{cases}, \tag{2}$$

where $r$ is the scale ratio.

In order to take into account these relationships, the Froude similarity was applied to the examined experimental tests. Scientific community agrees with the theory that values obtained from small scale tests are acceptable if the Froude number of the simulated current is comparable with the real ones (Hubl et al, 2009; Longo, 2011; Canelli et al. 2012). The open question concerns which is the maximum Froude number to accept small scale results; some authors (Hubl et al, 2009) suggest that the maximum acceptable Froude number for debris flow simulated in laboratory is 3, but it is demonstrated that debris flow in nature can assume Froude numbers greater than this value (Costa, 1984). Furthermore, small Froude number means high velocity value and, simultaneously, high thickness (and vice versa) but this conditions doesn't satisfy the characteristic of the majority of Alpine debris flows which are characterized by high velocity (greater than 10 m/s) and



relatively shallow depths (ranging from 0.2 to 1.5 m). For these reasons the Author decided to normalize force values respect to the hydrostatic force of the current and compared all the results with the corresponding Froude number.

## 3 Experimental setup and measuring procedure

Experiments were performed in a steel flume 4 meter long and 0.39 meter wide, in which a rigid barrier was positioned
orthogonally at the channel bottom. The slope is variable between 30° and 35°. The flow was started by the sudden emptying of a hopper into the flume (Fig. 1).

Four ultrasonic level measurers were mounted along the centre line of the channel at a known distance, decreasing progressively near the barrier. These devices had an acquisition frequency of 1 kHz and were used to evaluate both flow height and impact velocity on the barrier. To measure the normal thrust acting on the barrier, four load cells were installed at
the plate vertices.

Flow velocity at the barrier was estimated as the ratio of the distance between last ultrasonic level and the barrier location, and the difference between the time of first arrival of the front flow and the time of impact at the barrier. To observe the trend of the flow rate in the flume, the velocity values were evaluated at each sensors interval.

To control the evolution of the impact load at the barrier, a contact surface pressure gauge was used. This device, called
Tactilus®, is produced by Sensor Products LLC, and is designed to display a picture of the pressure distribution, measure and calculate min/max pressure, generate 2D and 3D modelling and region of interest viewing. It is made by a matrix of 32x32 piezoresistive sensors and allowed us to capture and record pressure conditions with a sampling frequency of 50 Hz (Fig. 2). To limit the possible formation of layers of air between the gauge and the barrier during the impact and to prevent to overestimate the impact load (Bagnold, 1939), the sensor perfectly adhered to the structure. This system is very useful to
understand the behaviour of the flow during the impact because it allows to verify the zones mainly stressed and confirms the hypothesis made on the determination of the peak impact force. In fact, observing Fig. 2, the pressure distribution assumes the typical triangular shape but the pressure values are greater than those expected: this confirms the hypothesis of a dominant dynamic component. Furthermore the Tactilus® allows to check the force values measured using load cells, with the advantage that in every points of the barriers it is possible to know the corresponding instant load values.

The tests were performed using saturated sand. The main characteristics of the material are listed in Table 1 and its grain-size distributions is shown in Fig. 3. The choice to use sand as the mixture material was made to obtain and easily check the characteristics of the flow. It is true that the grain size distribution used is not exhaustive and representative of a real debris flow (that has a very wide range of grain sizes), but the Authors want to avoid, at this stage of the study, the formation of over pressures due to the impact of boulders and their interactions inside the mixture. Furthermore, there is the necessity to
consider a homogeneous fluid scheme to evaluate the peak thrust.

However, to verify that the simulated currents could be assimilated to a debris flows, the six dimensionless parameters recommended by Iverson's theory (Iverson, 1997) were calculated (Table 1). Obviously the estimated values are referred to



the initial conditions. This is a simplification but it is possible to consider that the Bagnold Number, Darcy Number and Savage Number don't vary considerably during the flow and so, if these values fall into the debris flow region obtained from Iverson's theory, the mixture can be considered as a debris flow.

In this first stage of the study, only rigid and waterproof barrier was used, in order to reduce the possible deformations and consequently to correctly evaluate the force and better understand the dynamics of the impact.

## 4 Analytical approach

Several models were hypothesized to estimate the impact force of debris flow against rigid barrier. In particular, the impact force can be proportional either to hydrostatic pressure or kinetics flow height. Thus, three groups of relations can be used: hydro-static, hydro-dynamic and mixed models.

The equations referred to the first group have the following aspect:

$$F_{peak} = k \cdot \rho_m \cdot g \cdot h_f \cdot A \qquad (3)$$

where $F_{peak}$ is the maximum impact thrust in N, $k$ is an empirical coefficient, $\rho_m$ is the mean density of the debris impacting fluid in kg m$^{-3}$, $g$ is gravity in m s$^{-2}$, $h_f$ is the flow height in m and $A$ is the impact surface in m$^2$.

This formula is very popular because only requires debris density and flow height and usually flow height is considered equal to channel depth. The only limit is represented by $k$ factor that can assume values ranging from 2.5 to 11 (Lichtenhahn, 1973; Armanini, 1997; Scotton e Deganutti, 1997).

Hydro-dynamic models derive from the application of the momentum balance of the thrust under the hypothesis of homogeneous fluid; impact force can be evaluated as follows:

$$F_{peak} = \alpha \cdot \rho_m \cdot v_f^2 \cdot A \qquad (4)$$

where $\alpha$ is a dynamic coefficient and $v_f$ is the flow velocity in m s$^{-1}$.

The dynamic coefficient is the key point of this relation; it depends on the flow type, on the formation of a vertical jet-like wave during the impact and on the barrier type (Canelli *et al.*, 2012). In scientific literature, there is a wide range of proposed values for dynamic coefficient: Hungr and Kellerhals (1984) propose $\alpha$ equal to 1.5, Daido (1993) suggests values varying between 5 and 12, Zhang (1993) recommends a range between 3 and 5, Bugnion (2011) hypothesises value from 0.4 to 0.8, Canelli et al. (2012) between 1.5 and 5. From the values listed above it is clear that the range of variation of dynamic coefficient (between 0.4 and 12) deeply conditions the flow peak force and consequently the design of protection structures.

Furthermore, there are others formulations derived from hydro-dynamic relation. Hubl and Holzinger (2003) relate the Froude number (Fr) to normalised impact force and provide the following expression:

$$F_{peak} = 5 \cdot \rho_m \cdot v_f^{0.8} \cdot (g \cdot h_f)^{0.6} \cdot A \qquad (5)$$

Lamberti and Zanuttigh (2004), considering the total reflection of a current against a vertical wall and imposing the dynamic equilibrium, propose the relation:





$$F_{peak} = C_c \cdot \frac{\left(1 + \sqrt{2}F_r\right)^2}{2} \cdot \rho_m \cdot g \cdot h_f \cdot A \tag{6}$$

where *Cc* is an empirical coefficient calibrated considering the vertical acceleration caused by the presence of fine part and boulder.

Another equation to evaluate the dynamic impact of a debris flow against a vertical wall is presented by Armanini et al. (2011):

$$\tilde{F}_{peak} = \left(1 + \frac{1}{2} \cdot F_r^2\right) \cdot \left(1 + \frac{\alpha \cdot F_r^2}{1 + \frac{1}{2} \cdot F_r^2}\right) \tag{7}$$

where *α* is a coefficient equal to 1.

The mixed models consider both the hydro-static and the hydro-dynamic effects (Cascini et al, 2000; Arattano e Franzi, 2003; Brighenti et al., 2013); the general equation is:

$$F_{peak} = \frac{1}{2} \cdot \rho_m \cdot g \cdot h_f \cdot A + \rho_m \cdot v^2 \cdot A \tag{8}$$

Lately Jiang and Zhao (2015) have proposed a new formulation for impact force estimation, introducing the influence of the tangential forces during the impact due to the friction between flow and wall.

Combining the data obtained using the surface pressure gauge and flow characteristics (depositional height and velocity), we propose the following equation to estimate impact force on a rigid wall:

$$F_{peak} = F_{stat} + F_{dyn} \pm F_{drag} = \frac{1}{2} \cdot \rho_m \cdot g \cdot K_a \cdot (H_{max}^2 - h_f^2) \cdot B \cdot \cos\theta + \alpha \cdot \rho_m \cdot v_f^2 \cdot A \cdot \cos\beta - \rho_m \cdot g \cdot h_f \cdot \tan\varphi' \cdot$$
$$\frac{H_{max} - h_f}{\sin\theta} \cdot \cos\beta \cdot \cos\theta \cdot B \tag{9}$$

where $F_{stat}$ is the active earth force, $F_{dyn}$ is the dynamic force, $F_{drag}$ is the drag force (all the forces are evaluated in N), $K_a$ is active lateral earth pressure coefficient derived from Rankine theory, *θ* is slope angle in deg, *β* is the angle between the barrier and the normal at channel bottom, measured in deg, and $H_{max}$ is the maximum filling height behind the barrier in m (see Fig. 4).

Since static, dynamic and drag force do not reach their maximum value at the same time during the debris flow impact, $H_{max}$ should be considered equal to the height of the barrier $H_B$ in order to obtain the peak load. In this way, the static force reaches its maximum value.

The sign of the drag force depends if the current overflows or not the barrier. If there is overflow, the sign of the drag force is positive because it induces a deformation at the top of the barrier; vice versa the sign is negative because the flow produces a friction with the deposited material that reduces the dynamic effects.

This formulation contains both the intrinsic material characteristics, represented by static internal friction angle *φ* and density $\rho_m$, and flow conditions, depicted by flow height $h_f$ and current velocity $v_f$; moreover, the shape barrier, in terms of height $H_B$ and width *B*, and channel inclination are considered. In particular, including the shape of the barrier in the force peak calculation, the formula furnishes an innovative approach, since equations listed above are referred to flow conditions.




Furthermore, in these formulations there are no references at the channel condition in terms of inclination and dimension, while the equation proposed introduces these parameters.

The estimation of the static internal friction angle was done using the tilting box method (Burkalow, 1945); moreover, to verify the value obtained, a back analysis was done deriving the internal friction angle from the static force measured by

pressure device.

No measurements of bulk density variation were carried out during the impact phase; this value was hypothesized constant according to the theory of incompressible fluid.

Flow height and velocity were obtained using ultrasonic devices.

In order to follow the scale principles described in Sect. 2, Eq. (9) has been normalized by the hydro-static force relative to

the impacting front, obtaining

$$\tilde{F}_{peak} = \frac{F_{peak}}{\rho_m \cdot g \cdot h_f \cdot A} = \frac{1}{2} \cdot K_a \cdot (n^2 - 1) \cdot \cos\theta + \alpha \cdot F_r^2 \cdot \cos\beta - \tan\varphi' \cdot \frac{n-1}{\sin\theta} \cdot \cos\beta \cdot \cos\theta \qquad (10)$$

where $n$ is the filling ratio and $F_r$ the Froude number of the current.

About the filling ratio, it is the ratio between the maximum filling height behind the barrier and the flow height; this number allows to relate flow thickness with barrier dimension.

When n is equal to 1, the dimensionless force is reduced to Eq. (4). This means that if the barrier is hit by a volume moderately small, composed by only one surge, the peak force is totally governed by dynamic component.

**5 Validation of the proposed model**

Analysing the trend of the total impact force in time (Fig. 5), the hypothesized model is confirmed: in fact, it is possible to highlight how the peak force acting on the barrier can be assumed as the sum of two components: one in which static

behaviour is predominant and one in which dynamic effects, due to the formation of a vertical jet-like wave, contribute to generate peak. Furthermore, observing the behaviour of the flow in time, the succession of static and dynamic force is justified because the mobilized volume hits against barrier with consecutive surges.

Fig. 6 shows the trend of the normalized force measured, $\tilde{F}$, in function of the Froude number, $Fr$, for the two different channel inclinations (30° and 35°). The laboratory data have been compared with the equation proposed by Hungr and

Kellerhals (1984), Armanini and Scotton (1992), Cascini et al. (2000), Hubl and Holzinger (2003), Zanuttigh and Lamberti (2004) and Armanini et al. (2011).

Most of the experimental data fall between the values estimated using Hubl and Holzinger's equation (2003) and Hungr and Kellerhals's equation (1984) with dynamic coefficient equal to 1.5.

In Eq. (10), the only parameter unknown is the dynamic coefficient α because n can be hypothesized on the basis of the

barrier height and the estimated flow thickness.



Fig. 7 represents the trend of the proposed equation compared with the experimental data for different inclination of the flume. In particular it is possible to notice that the major part of the data fall into a region defined by an upper and a lower limit, evaluated respectively using the proposed equation with dynamic coefficients equal to 1.2 and 0.5.

The difference between Fig. 7(*a*) and Fig. 7(*b*) is the value of filling ratio, respectively equal to 11 and 9. The fact that the
filling ratio is greater when inclination is greater supports the hypothesis that n is directly related with velocity flow. In fact, a correlation between height flow and velocity has been observed in the laboratory test analysis. Fig. 8 clearly shows this correlation: a linear dependence exists between filling ratio (that stores thickness information) and Froude number (that stores velocity information).

According to these observations, the Authors want to focus on the trend of the proposed equation: for small Froude number
values, relating to the other analyzed formulations, it is evident how the static component is predominant compared with the dynamic one. On the other hand, for high values of Froude number the equation is close to the hydro-dynamic models. This means that if the current has small velocity and, therefore, higher flow height, the peak impact force presents a hydro-static behaviour; opposite, with high velocity values and small thickness, the hydro-dynamic components is relevant and it provides the major contribute to estimation of impact thrust.

About the variation of dynamic coefficient, this is extremely influenced by the formation of the vertical jet like wave. The fact that α is not so higher than the unity, confirms the goodness of experimental tests; moreover, it suggests that the filling of the barrier occurs for a succession of surges and the peak force is not influenced by overpressure due to reflected waves.

**6 Conclusion**

This study has the aim of reviewing the dynamics of debris flow impact against rigid structures and providing a new simple
formulation to predict peak thrust.

The equation proposed differs from other formulations because takes into account both flow characteristics and material properties and barrier dimensions. It could easily be used to safely design protection barriers, considering the filling ratio as the ratio between barrier height and flow thickness.

The model developed has a good capability to predict force measured during the laboratory tests. Further studies should be
done to verify and, if necessary, to adjust this equation comparing with data obtained from real case events.

**Acknowledgements**

The authors would like to acknowledge the Consorzio Triveneto Rocciatori (CTR) for its financial support of part of this project.



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



**Tables**

Table 1 - Main initial properties of the mixture used.

| Mixture Main Initial Properties | | |
|---|---|---|
| Friction angle | $\varphi'$ [°] | 29 |
| Density of the grain | $\rho_s$ [kg/m$^3$] | 2630 |
| Density of the flow | $\rho_f$ [kg/m$^3$] | 1920 |
| Solid volume fraction | $C_s$ | 0.6 |
| Fluid volume fraction | $C_f$ | 0.4 |
| Savage number | $N_{Sav}$ | 0.144 |
| Bagnold number | $N_{Bag}$ | 888 |
| Mass number | $N_{Mass}$ | 3.75 |
| Darcy number | $N_{Dar}$ | 576 |
| Reynolds number | $N_{Rey}$ | 236 |
| Friction number | $N_{Fric}$ | 6628 |



**Figures**

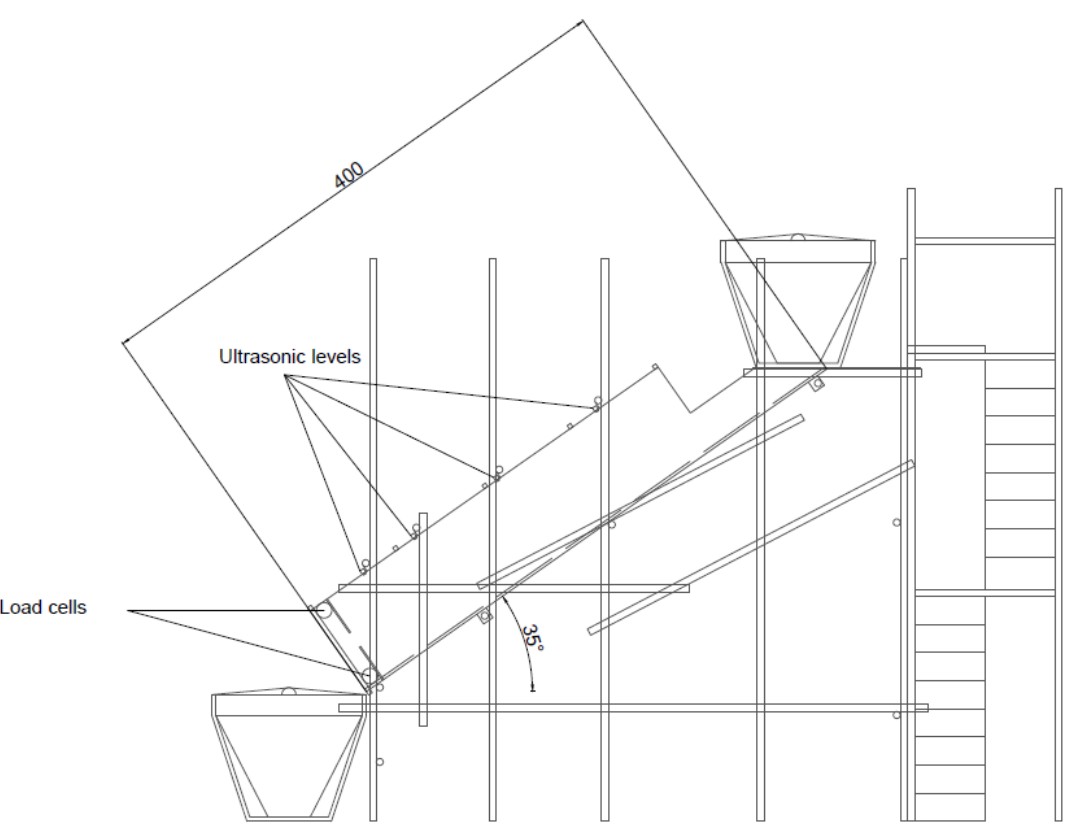

**Figure 1: Scheme of the flume and of the starting mechanism.**





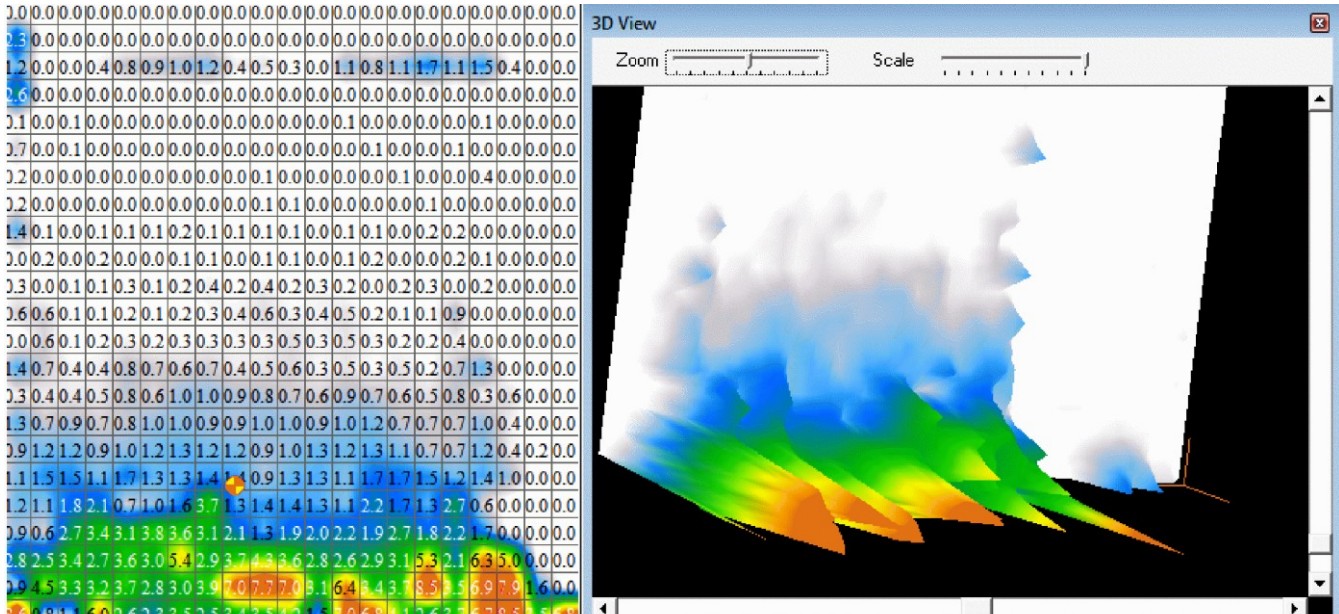

**Figure 2: 2D and 3D impact representation registered by Tactilus® pressure sensor.**






Figure 3: Grain size distribution of the mixture.



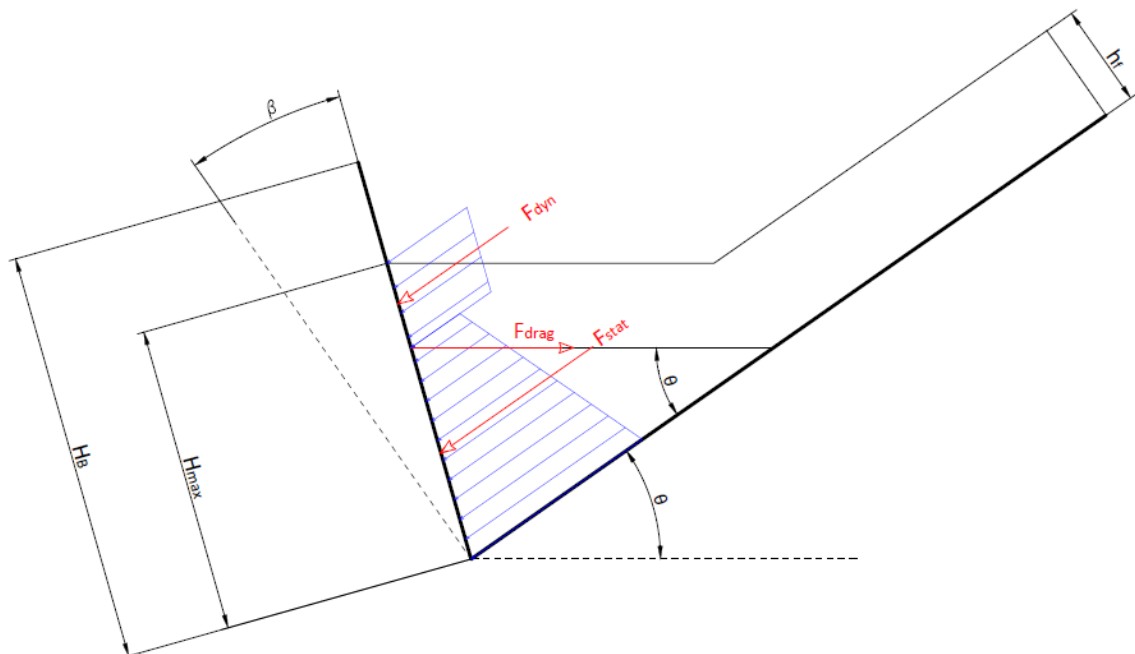

**Figure 4: Scheme of flow impact and assumed filling process for the calculation of dynamic, static and drag load on the rigid barrier.**





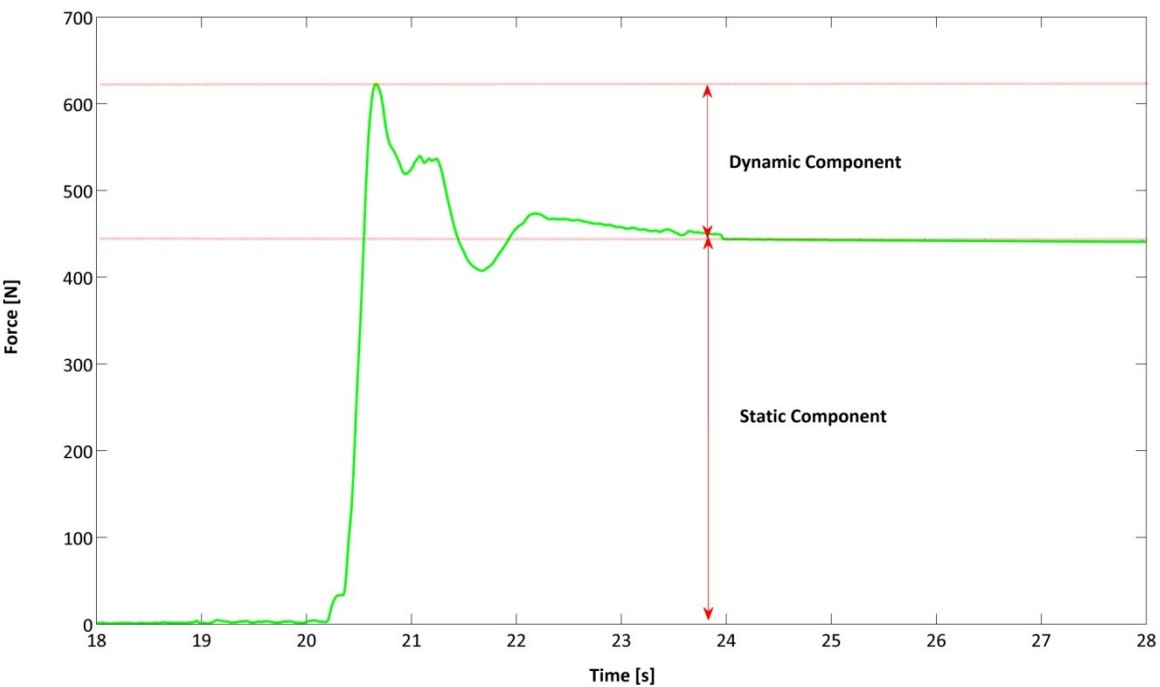

**Figure 5: Total impact force measured at load cells vs time; the static and the dynamic component are highlighted.**

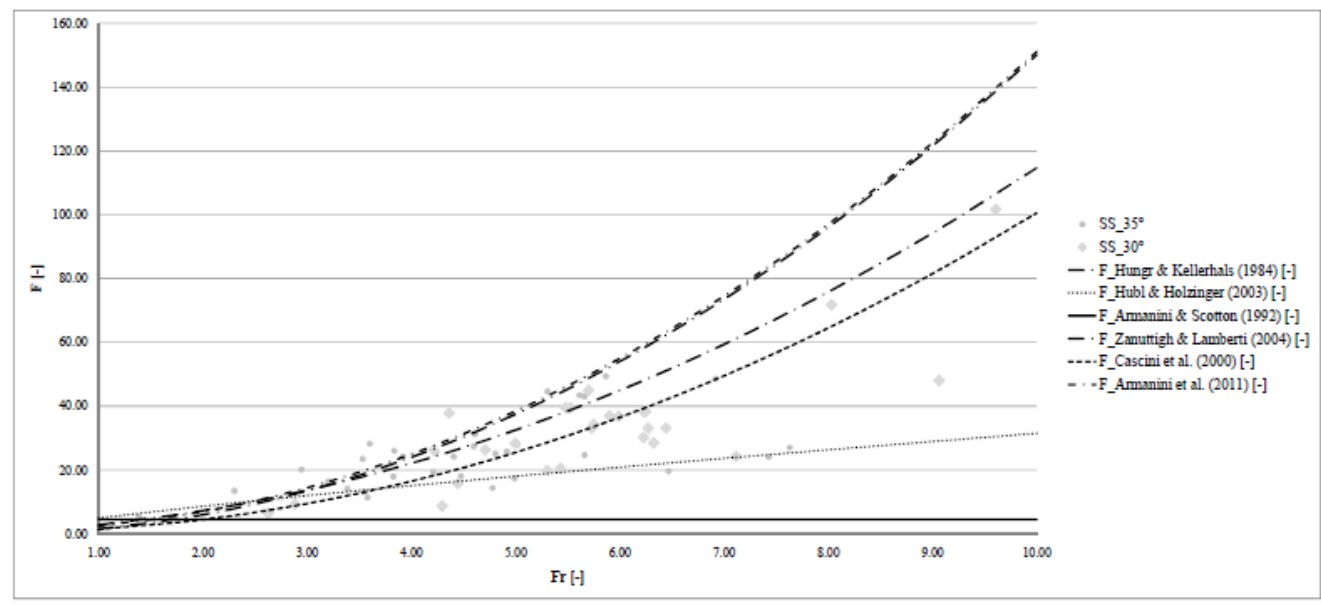

**Figure 6: Trend of normalized force measured (points) and predicting model (line) in function of the Froude number. The labels**
5 **SS_30° and SS_35° correspond to force values evaluated using saturated sand with an inclination of the flume of 30° (circle points) and 35° (diamond points) respectively.**




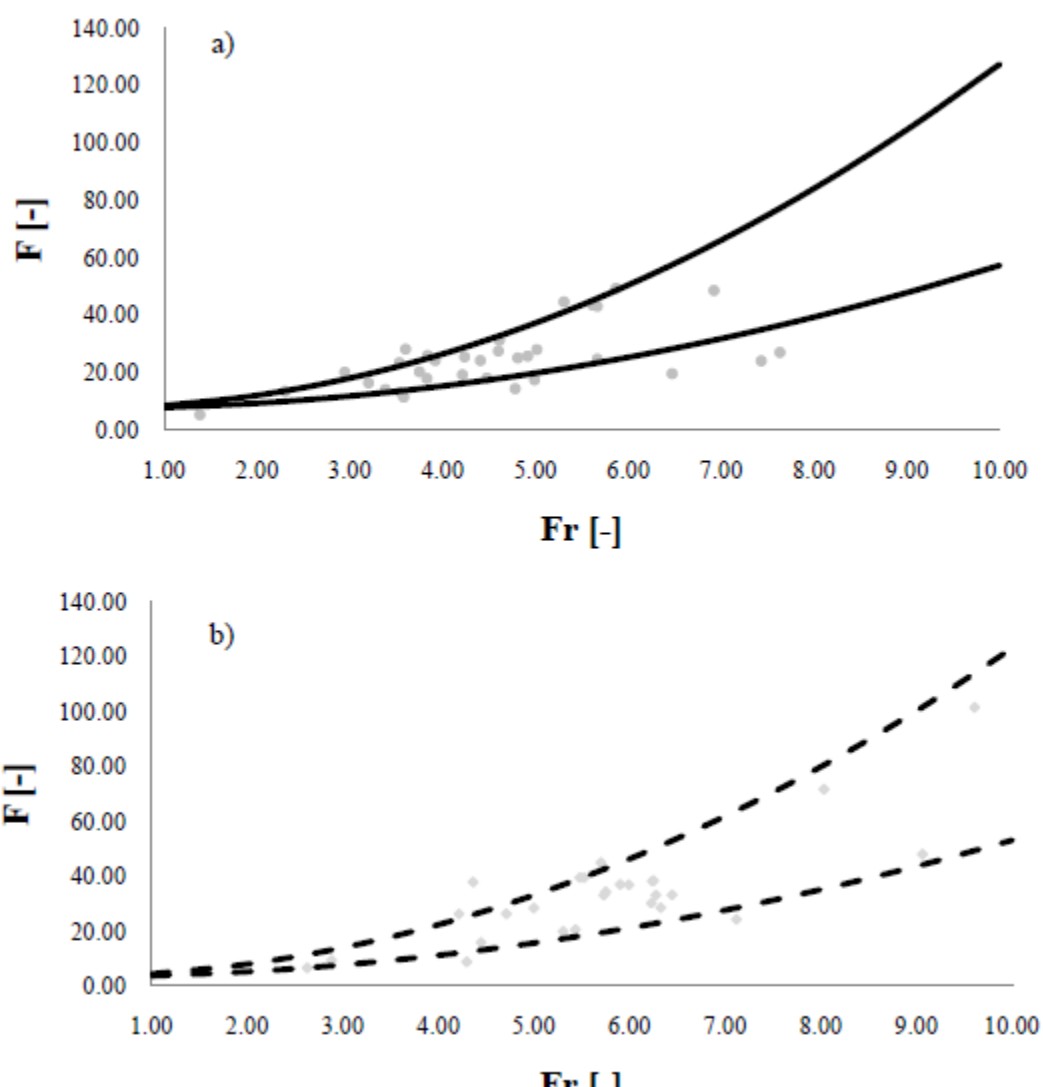

**Figure 7: Dimensionless force vs Froude number for flume inclinations equals to 30° (7a) and 35° (7b): the points fall in a region derived by the proposed model using α = 1.2 (upper limit) and 0.5 (lower limit). The two regions are obtained using n = 9 (7a) and n = 11 (7b).**




Figure 8: Linear correlation between filling ratio, n and Froude number, Fr for dataset obtained respectively by flume inclination equal to 30° (8a) and 35° (8b).