# Peer review of "Debris flow impact estimation on a rigid barrier"

_Natural Hazards and Earth System Sciences, 2016_

## Short Comment (SC1) · 31 Mar 2016

Dear Authors the paper deals with a interesting and important problem. The proposed work is based on experimental testing in a channel where several test have been performed to analyse the influence of the flow nature on the impact. In order to use these results on a real scale problem scale effect have to taken into account. A further discussion on this point would be appreciate since it can been seen as the most weak point of a sound and well settle research.

---

## Referee Comment (RC1) · Anonymous Referee #1 · 15 Apr 2016

This paper deals with the study of the impact of a debris flow on a rigid barrier by means of experimental and theoretical evaluations. Particularly interesting are the experimental results obtained with an advanced device able to measure the induce pressure on a rigide obstacle. This kind of measure is very new and original and can help in solving an important issue in this field: the determination of the pressure rate. The paper is well written and understandable but a language check by a mother tongue speaker is strongly recommended.

---

## Author Comment (AC1) · 22 Apr 2016

Thank you very much Prof. Ferrero for your comment. Obviously the scale effect is a key point in experimental tests; starting from the dimensional analysis of the problem, we decided to apply Froude similarity to our laboratory results. In particular, we compared the Froude number of each test with those of real debris flow events described in literature. As we wrote in the paper, a discussion on the limit scale values (e.g. maximum acceptable Froude number) is necessary in order to come to universal recognized scale principles. Definitively we totally agree with your comment.

---

## Author Comment (AC2) · 22 Apr 2016

Thank you very much for your comments. The Authors are satisfied that the paper has been appreciated by this referee. The barrier design for the protection against debris flow risk is still an open issue; this work is a first approach to reduce the lack of knowledge relate to these phenomena. We think that the new formulation proposed could furnish good results for flexible barrier: further studies are needed to cover this type of structures since they are widely used. Furthermore, due to the simplicity of the formulation, this approach should be implemented in numerical code in order to create a useful tool for the designer. Concerning the quality of the language, the paper will be sent to a mother tongue speaker for a substantial check.

[Figure]

2016.

---

## Referee Comment (RC2) · Anonymous Referee #2 · 25 Apr 2016

In this paper, authors revised the impact behavior of debris flow on a rigid barrier and proposed a new formulation for the evaluation of the peak force. Furthermore an interesting tool for the measurement of the evolution of impact pressure in time is described.

The formulation is derived from simple geometrical assessment; the innovation respect to the others equations shown in literature is represented by introduction of barrier dimensions and channel characteristics. It seems to be very easy to use this formula and the presented graphs show a significant correspondence between the theoretical and experimental data. But, as the other equations, it is extremely influenced by the dynamic coefficient (alpha); can the authors add more details about this parameter?

The pressure sensor used in the experimental test seems very a very useful tool to understand the behavior of the flow during impact. I suggest to add more information

about it.

Even if the paper is well written, for the publication the English should be improved.

For all these reasons the paper should be considered by this journal after minor revision.

---

## Author Comment (AC3) · 17 May 2016

Thank you very much for spending time reviewing this manuscript and providing useful comments. Your comments and critical observations are very informative and constructive. More specific replies to all comments are now given.

1 - [...] it is extremely influenced by the dynamic coefficient (alpha); can the authors add more details about this parameter?

The following paragraph has been added at page 4, line 26: "In particular, the drainage capability of the barrier reduces the magnitude of this coefficient due to the rapid discharge of the fluid portion through the barrier, preventing the formation of wave overpressure. Another aspect to take into account while choosing $\alpha$ is the grain size distribution of the debris flow: if it is predominantly coarse, the dynamic coefficient is greater

since there is a local overpressure build up due to the impact of single boulders on the barrier."

2 - The pressure sensor used in the experimental test seems a very useful tool to understand the behavior of the flow during impact. I suggest to add more information about it.

The following paragraph has been added at page 3, line 24: "In the experimental tests, this device was also used to verify the occurrence of vertical wave overpressure. The capability to record impact pressure in real time allows to understand and to detect the most stressed zones of the barrier. In this way, it is possible to verify the accuracy of the hypotheses done about the behaviour of the current during the impact."

3 - Even if the paper is well written, for the publication the English should be improved.

The English has been checked and improved. You can find the revised manuscript attached as supplement.

Please also note the supplement to this comment:
http://www.nat-hazards-earth-syst-sci-discuss.net/nhess-2016-80/nhess-2016-80-AC3-supplement.pdf

**Supplement:**

[revised manuscript text omitted]
. In the experimental tests, this device was also used to verify the occurrence of vertical wave overpressure. The capability to record impact pressure in real time allows to understand and to detect the most stressed zones of the barrier. In this way, it is possible to verify the accuracy of the hypotheses done about the behaviour of the current during the impact.

The tests were performed using saturated sand. The main characteristics of the material are listed in Table 1 and its grain-size distributions is shown in Fig. 3. The choice to use sand as the mixture material was made to obtain and easily check the characteristics of the flow. It is well known that the grain size distribution used is not exhaustive and representative of a real debris flow (which is generally made up of a very wide range of grain sizes) but the Authors wanted to avoid, at this stage of

the study, the formation of over pressures due to the impact of boulders and their interactions inside the mixture. Furthermore, there is the necessity to consider a homogeneous fluid scheme to evaluate the peak thrust.

However, to verify that the simulated currents could be assimilated to debris flows, the six dimensionless parameters recommended by Iverson's theory (Iverson, 1997) were calculated (Table 1). Obviously, the estimated values are referred to the initial conditions. This is a simplification, but it is possible to consider that the Bagnold Number, Darcy Number and Savage Number don't vary considerably during the flow. Therefore, when these values fall into the debris flow region obtained from Iverson's theory, the mixture can be considered as a debris flow.

In this first stage of the study, only rigid and waterproof barrier was used, in order to reduce the possible deformations and consequently to correctly evaluate the force and better understand the dynamics of the impact.

**4 Analytical approach**

Several models were hypothesized to estimate the impact force of debris flow against rigid barrier. In particular, the impact force can be proportional either to hydrostatic pressure or kinetics flow height. Thus, three groups of relations can be used: hydro-static, hydro-dynamic and mixed models.

The equations referred to the first group have the following aspect:

$$F_{peak} = k \cdot \rho_m \cdot g \cdot h_f \cdot A \tag{3}$$

where $F_{peak}$ is the maximum impact thrust in N, $k$ is an empirical coefficient, $\rho_m$ is the mean density of the debris impacting fluid in kg m$^{-3}$, $g$ is gravity in m s$^{-2}$, $h_f$ is the flow height in m and $A$ is the impact surface in m$^2$.

This formula is very popular because it only requires debris density and flow height and usually flow height is considered equal to channel depth. The only limit is represented by $k$ factor that can assume values ranging from 2.5 to 11 (Lichtenhahn, 1973; Armanini, 1997; Scotton e Deganutti, 1997).

Hydro-dynamic models derive from the application of the momentum balance of the thrust under the hypothesis of homogeneous fluid; impact force can be evaluated as follows:

$$F_{peak} = \alpha \cdot \rho_m \cdot v_f^2 \cdot A \tag{4}$$

where $\alpha$ is a dynamic coefficient and $v_f$ is the flow velocity in m s$^{-1}$.

The dynamic coefficient is the key point of this relation; it depends on the flow type, on the formation of a vertical jet-like wave during the impact and on the barrier type (Canelli *et al.*, 2012). In particular, the drainage capability of the barrier reduces the magnitude of this coefficient due to the rapid discharge of the fluid portion through the barrier, preventing the formation of wave overpressure. Another aspect to take into account while choosing α is the grain size distribution of the debris flow: if it is predominantly coarse, the dynamic coefficient is greater since there is a local overpressure build up due to the impact of single boulders on the barrier.

[revised manuscript text omitted]